# Effect of Al and La Doping on the Structure and Magnetostrictive Properties of Fe_73_Ga_27_ Alloy

**DOI:** 10.3390/ma16010012

**Published:** 2022-12-20

**Authors:** Jinchao Du, Pei Gong, Xiao Li, Shaoqi Ning, Wei Song, Yuan Wang, Hongbo Hao

**Affiliations:** 1School of Materials Science and Engineering, Inner Mongolia University of Technology, Hohhot 010051, China; 2Baotou Rare Earth Research Institute, State Key Laboratory of Baiyunebo Rare Earth Resources Research and Comprehensive Utilization, Baotou 014010, China; 3Inner Mongolia Autonomous Region Strategic Research Center for Science and Technology, Hohhot 010051, China

**Keywords:** Al addition, La addition, microstructure, microhardness, magnetostriction

## Abstract

The changes of microstructure, magnetostriction properties and hardness of the Fe_73_Ga_27−x_Al_x_ alloy and (Fe_73_Ga_27−x_Al_x_)_99.9_La_0.1_ alloy (x = 0, 0.5, 1.5, 2.5, 3.5, 4.5) were studied by doping Al into the Fe_73_Ga_27_ and (Fe_73_Ga_27_)_99.9_La_0.1_ alloy, respectively. The results show that both the Fe_73_Ga_27−x_Al_x_ alloy and (Fe_73_Ga_27−x_Al_x_)_99.9_La_0.1_ alloy are dominated by the A2 phase, and the alloy grains are obvious columnar crystals with certain orientations along the water-cooled direction. A proportion of Al atoms replaced Ga atoms, which changed the lattice constant of the alloy, caused lattice distortion, and produced vacancy effects which affected the magnetostriction properties. La atoms were difficult to dissolve in the matrix alloy which made the alloy grains smaller and enhanced the orientation along the (100) direction, resulting in greater magneto-crystalline anisotropy and greater tetragonal distortion, which is conducive to improving the magnetostriction properties. Fe_73_Ga_24.5_Al_2.5_ alloy has a saturation magnetostrictive strain of 74 ppm and a hardness value of 268.064 HV, taking into account the advantages of saturated magnetostrictive strain and high hardness. The maximum saturation magnetostrictive strain of the (Fe_73_Ga_24.5_Al_2.5_)_99.9_La_0.1_ alloy is 115 ppm and the hardness is 278.096 HV, indicating that trace La doping can improve the magnetostriction properties and deformation resistance of Fe-Ga alloy, which provides a new design idea for the Fe-Ga alloy, broadening their application in the field of practical production.

## 1. Introduction

Due to the advantages of high energy density, fast response speed, high Curie temperature and excellent magnetostriction properties, the Tb-Dy-Fe alloy was widely used in sensors, brakes, transducers, micro displacement drivers and shock absorbers in the 1970s [1,2,3,4], but its large saturation magnetic field and poor mechanical performance limited its extensive application [5]. As a new type of magnetostrictive material developed after Tb-Dy-Fe, the Fe-Ga alloy has the advantages of a low saturated magnetic field, good mechanical properties and low material cost. Although the saturation magnetostrictive strain of a single crystal Fe-Ga alloy along the (100) easy magnetization direction can reach up to 400 ppm [6], its wide utilization is limited by its complex preparation process, poor mechanical properties at room temperature, high cost and size limitation [7]. The saturation magnetostrictive coefficient of the polycrystalline Fe-Ga alloy is low. In order to improve its magnetostriction properties, researchers have carried out extensive research on element doping, alloy preparation processes, heat treatment processes and pre-compression stress treatment. As an effective and simple method of improving the magnetostriction properties, element doping has attracted the attention of many magnetic researchers.

The Al and Ga elements belong to the same main group with similar atomic radii, and the Al atom can replace the Ga atom to solve the Fe (Ga) matrix. Studies have shown that the saturation magnetostrictive strain does not decrease significantly when Al doping is less than 10% in an Fe-Ga alloy, and Al has the advantage of having good ductility, processability and low price. Li Hua [8] studied the Fe_81_Ga_16.5_Al_3.5_ alloy and its saturation magnetostrictive strain is 202 ppm. Srisukhumboworncha [9] studied the directionally grown Fe_80_Ga_15_Al_5_ alloy and found that its saturation magnetostrictive strain reached 234 ppm. Han Zhiyong [10] found that the saturation magnetostrictive strain of the Fe_87_Ga_13_ polycrystalline alloy reached 286 ppm under a 20 Mpa preload, which was 34.27% higher than the saturation magnetostriction under a zero-preload stress.

By reason of the 4f electron layer and strong spin–orbit effect of rare earth elements, the Fe-Ga alloy A2 phase is preferentially oriented along the (100) direction, contributing to greater magneto-crystalline anisotropy, thus heightening the magnetostriction coefficient. The saturation magnetostrictive value of the Fe_83_Ga_17_Tb_0.5_ alloy is 149 ppm, as measured by Yu Quangong [11]. Jiang Liping [12] found that the saturation magnetostrictive value of the Fe_83_Ga_17_Dy_0.2_ alloy reached 300 ppm. Meng Chongzheng [13] prepared the (Fe_83_Ga_17_)_99.9_Tb_0.1_ alloy by directional solidification and its saturation magnetostrictive value clocked up 255 pm. Yao Zhanquan [14] measured that the saturation magnetostrictive value of the Fe_83_Ga_17_Ce_0.8_ alloy was up to 356 ppm and discovered the second phase CeGa_2_ distributing in a network at the grain boundary of Fe-Ga solid solution phase. All these studies demonstrate that rare earth doping can effectively enhance the magnetostrictive coefficient of the alloy.

As a light rare earth element with the lowest atomic number, La lacks the 4f electron layer and has a price that is more competitive than heavy rare-earth elements. As the Fe-Ga binary phase diagram shows, the saturation magnetostrictive value of the Fe_100−x_Ga_x_ alloy presents two peak values at x = 17~19 and x = 27~29 [15]. In this paper, the Fe_73_Ga_27_ and (Fe_73_Ga_27_)_99.9_La_0.1_ alloys are selected as the matrices to be doped with Al element, respectively, and their effects on the saturation magnetostrictive strain and hardness are studied.

## 2. Materials and Methods

Fe, Ga, Al, La metal elements with purity greater than 99.95% were selected as raw materials and 70 g of raw materials were weighed using an electronic balance. The circulating water cooler was started before smelting. Firstly, the vacuum pressure was kept below 10 Pa by using a mechanical pump. The molecular pump was then turned on, eventually making the vacuum degree under 5 × 10^−3^ Pa and aerating with high purity argon as the protective gas. During alloy smelting, the arc current was maintained at 50–400 A. After the sample was completely cooled under the action of circulating water, the sample was turned over and smelted repeatedly three times to obtain a sample with uniform composition and good shape.

An AxioImagerAim metallographic microscope was used to observe the structure and grain orientation of alloy grain boundaries and matrices. An X-ray Diffractometer (XRD) of X-PertPro type was used to analyze lattice constants and determine phase compositions of alloys and the Sima500 scanning electron microscope produced by Carl Zeiss of Germany was utilized to observe the microstructure of the alloy and clarify the composition of different areas of the alloy through its own energy spectrometer. The magnetostrictive values of the alloys were measured using the resistance strain gauge method and the microhardness of the alloys were gauged with a MH-5AC microhardness tester.

## 3. Results

### 3.1. XRD Analysis

Figure 1 shows the X-ray diffraction patterns of the Fe_73_Ga_27−x_Al_x_ and Fe_73_Ga_27−x_Al_x_)_99.9_La_0.1_ alloys. The test angle ranges 20–100° and the step length is 0.02°/min. From Figure 1, the characteristic peak of the alloy remains unchanged after doping with Al and La and it is still composed of three diffraction peaks (110), (200) and (211). The alloy phase is mainly composed of the A2 phase. Compared with the standard diffraction spectrum of bcc Fe, the characteristic peaks of the Fe_73_Ga_27−x_Al_x_ and Fe_73_Ga_27−x_Al_x_)_99.9_La_0.1_ alloys appear to have a small angle deviation, which is related to Al atoms substituting Ga atoms in the Fe (Ga) solid solution [16]. Although Al and Ga belong to the same main group, they dissolved well with each other without a second phase requiring to be generated. The radius of the Al atom (1.43 Å) was larger than the Ga atom, (1.40 Å), when the Al atom replaced the Ga atom in the solid solution lattice resulting in distortion in the alloy lattice, therefore showing a small angle shift of the diffraction peak [17]. Despite no new diffraction peak generating in the alloy diffraction spectrum after doping with La, the intensity of the (200) diffraction peak showed obvious change. On the basis of the following paragraph, when the I_200/_I_110_ reached a maximum value of 140.77%, the (200) diffraction peak became the main peak, indicating that La doping enhanced the (100) orientation of the alloy.

Table 1 is the lattice constant a(Å) and the relative intensity of the diffraction peak (I_200_/I_110_) obtained by calculating the XRD patterns of the Fe_73_Ga_27−x_Al_x_ and (Fe_73_Ga_27−x_Al_x_)_99.9_La_0.1_ alloys. The second and third columns display the results of lattice constants (a) and I_200/_I_110_(%) of the Fe_73_Ga_27−x_Al_x_ alloys which change with Al content. In the Fe_73_Ga_27−x_Al_x_ alloy, the lattice constant first increases and then decreases following the Al content, reaching maximum at x = 2.5% and minimum at x = 1.5% several times. The I_200/_I_110_(%) of the alloy also shows a trend of increasing first and then decreasing, showing two inflection points appearing at x = 1.5% and 3.5%, respectively. The Al atoms gradually replace the Ga atoms in the alloy, resulting in the variation of lattice constant and lattice gap generating a vacancy effect [18], which affects the magnetostriction properties. According to the displacement solid solution theory in the basis of material science [19], when atoms with a large radius replace atoms with a small radius to form a solid solution, the lattice constant of the alloy will increase.

The fourth and fifth columns list the results of lattice constants a(Å) and I_200/_I_110_(%) of the (Fe_73_Ga_27−*x*_Al*_x_*)_99.9_La_0.1_ alloy with varying Al content. The lattice constant generally increases first and then decreases after doping with a small amount of La. However, the lattice constant values are all smaller than those of the Fe_73_Ga_27−x_Al_x_ alloy with the same Al composition, indicating that it is difficult for La to be dissolved in the matrix. The I_200_/I_110_ (%) of the alloy generally presents a trend of waveform change, attaining the maximum value of 140.77% at x = 2.5%. The increase in the I_200_/I_110_ ratio reflects that the preferred orientation of the alloy in the (100) direction is enhanced at this time. The (100) direction is the internal stress gradient direction of grain growth and the easy magnetization direction of the Fe-Ga alloy [20], which has a major impact on the magnetostrictive property of the alloy.

### 3.2. Analysis of Alloy Structure and Morphology

Figure 2 presents a metallographic photo of the Fe_73_Ga_27−x_Al_x_ alloy. It shows that the alloy structure is made up of a gray matrix phase and a black dot which disperses in the matrix and at the grain boundary. The Fe_73_Ga_22.5_Al_4.5_ alloy has visibly more black dots than other alloys. The grain shape of the alloy appears columnar and its orientation along the direction of water cooling is very clearly shown. As mentioned in the literature, the gray matrix phase structure is the Fe (Ga) solid solution phase with a bcc structure [21]. There are three reasons for the formation of black dot tissue. Firstly, Al atoms are not completely dissolved in the matrix alloy and the rest of the Al exists in the form of inclusions, growing into the black dots. Secondly, during solidification, a fraction of Ga atoms exists in the form of point precipitates to form a Ga-rich phase. Thirdly, it might be caused by matrix phase shedding in the process of grinding and polishing.

Figure 3 presents the metallographic photo of the (Fe_73_Ga_27−x_Al_x_)_99.9_La_0.1_ alloy. It can be seen from Figure 3 that, although the alloy is still dominated by columnar crystals with a certain orientation along the water-cooled direction, the microstructure morphology changes greatly. The black precipitates vary from point to sheet, spreading at the grain boundary. According to the following scanning photos and EDS composition analysis, La elements exist in the form of a precipitated phase, which is consistent with the literature [22]. The atomic radius of La (2.74 Å) is far larger than Ga and Al atoms and the rare earth element is difficult to dissolve in the matrix phase of the Fe-Ga alloy, resulting in the decrease in the lattice constant of the alloy and the role of refining the grains. As the literature references, the finer the metal grains are, the higher the strength and hardness are, and the better the plasticity and toughness [23]. It can be predicted that after adding La the hardness of the alloy will be greater than the alloy without La doping.

The Fe_73_Ga_27−x_Al_x_ alloy and (Fe_73_Ga_27−x_Al_x_)_99.9_La_0.1_ alloy at x = 0 % and x = 2.5% exhibited a preferred orientation in the (100) direction and, as shown in Figure 4, this composition point alloy has excellent magnetostriction properties and hardness. In order to further ascertain the composition of the matrix phase and precipitated phase, spot analysis and surface analysis were carried out using scanning electron microscopy and composition analysis of the selected area was carried out by the EDS. Figure 4a–d are SEM photos of the Fe_73_Ga_27−x_Al_x_ alloy (x = 0% and x = 2.5%), Figure 4e and Figure 4f are SEM photos of the (Fe_73_Ga_27−x_Al_x_)_99.9_La_0.1_ alloy (x = 0% and x = 2.5%). The grain boundary phases include A, D, G and J. Matrix phases include B, E, H and K. Precipitated phases include C, F, I and L.

Table 2 displays the EDS results in different areas of the Fe_73_Ga_27−x_Al_x_ and (Fe_73_Ga_27−x_Al_x_)_99.9_La_0.1_ alloys. Zone B is the matrix phase composition of the Fe_73_Ga_27_ alloy. EDS composition analysis shows that the atomic ratio of Fe to Ga is 73.87:26.13, which is close to the nominal composition, suggesting that the alloy is uniformly melted without component segregation. The H zone and K point are the matrix phase components of (Fe_73_Ga_27_)_99.9_La_0.1_ and (Fe_73_Ga_24.5_Al_2.5_)_99.9_La_0.1_, of which the relative content of La elements are 0.06%, which is far lower than the precipitates of I point (15.30%) and L point (23.75%), indicating that it is difficult for the La element to dissolve in the matrix. Zone E and point F are the matrix phase and precipitate phase of the Fe_73_Ga_24.5_Al_2.5_ alloy, respectively. The relative content of the Al element at point F (1.42%) is evidently higher than that of the Al element at zone E (0.72%), which evinces a part of the Al atoms in the alloy of this composition entering the matrix to replace Ga atoms. Additionally, because the radius of Al atoms is greater than that of Ga atoms, the lattice constant of the Fe_73_Ga_24.5_Al_2.5_ alloy will be reduced when solid solution is formed.

In order to determine the composition of the black dots in the Fe_73_Ga_27−x_Al_x_ (x = 0, 0.5, 1.5, 2.5, 3.5, 4.5) alloy, two points of each component were selected for EDS analysis under SEM. Figure 5 presents the SEM images of each component of the Fe_73_Ga_27−x_Al_x_ alloy and Table 3 displays the EDS component analysis results for each alloy.

According to the results in Table 3, the composition of the large proportion of black dots is similar to the composition of their respective matrix. Observing the points C, D and K, the relative contents of Al elements were 59.89, 60.45 and 55.68, respectively, which were higher than those of Fe and Ga, and a Ga-rich region was not observed in the alloy. We can therefore speculate that the black dots tissue in the alloy is more likely to be composed of the deciduous matrix phase and Al element in the form of inclusion.

### 3.3. Magnetostriction Properties

Figure 6a displays saturation magnetostrictive strain(λs) curve of the Fe_73_Ga_27−x_Al_x_ alloy changing with external magnetic field. As Figure 6a shows, the saturation magnetostrictive strain of the alloy rises with the increase in the external magnetic field. When the magnetic field strength reaches 1500Oe, the λs remains invariable with the external magnetic field, reflecting that it is saturated at this time. Figure 6b presents the curve of the λs for the Fe_73_Ga_27−x_Al_x_ alloy with Al content transformation. In Figure 6b, its λs curve varies within 58–76 ppm. The maximum value of the λs curve reaches 76 ppm at x = 0% and the λs of the alloys descend in the range of 0–1.5%. When the Al content in the alloy is low, Al atoms dissolve into the alloy instead of Ga atoms α-Fe, conducing the augmentation of the lattice constant, the attenuation of the vacancy effect, the increase in electron density around the alloy and the decrease in magnetostriction performance. The λs curve between x = 1.5% and 2.5% ascends slightly. After doping appropriate amounts of the Al element, the adjacent Ga-Ga atom pair in the (100) direction of the Fe-Ga alloy is destroyed [24], contributing to lattice distortion, forming short-range ordered clusters, and reducing the shear modulus of the alloy, which is conducive to improving the magnetostriction properties. As Al content continues to increase, the λs curve decreases within the range of x = 2.5–4.5%. When the addition of Al content becomes higher, the number of valence electrons transferred in the alloy will also rise, leading to an augmentation in the number of electrons on the secondary energy band. Only when the electrons on the secondary energy band are filled, will the remaining valence electrons be transferred to another magnetic energy band, resulting in a diminution in the magnetic moment of the Fe atom and the magnetostriction properties of the alloy [25].

Figure 7a shows the curve of the saturation magnetostriction strain of the (Fe_73_Ga_27−x_Al_x_)_99.9_La_0.1_ alloy alteration with the external magnetic field. According to Figure 6 and Figure 7a, the λs curves of the alloy increase with the addition of the external magnetic field. When the external magnetic field strength reaches 1600Oe, the λs curve remains at the saturated state. Figure 6b shows the curve of λs for the alloy changing with the Al content. After adding the La element, the λs of the alloys is kept within 73–115 ppm, which is meliorative in comparison to the Fe_73_Ga_27−x_Al alloy. In a range of x = 0–0.5%, the λs of the alloy diminishes due to the increase in the lattice constant. According to literature [26], the lattice constant of the alloy rises, the void in the lattice decreases, and the vacancy effect weakens, which is beneficial to the magnetostriction performance. Within x = 0.5–2.5%, the λs of the alloy rises and reaches the maximum value of 115 ppm at x = 2.5%. After adding the La element in the Fe-Ga alloy, large portions of the La exist in the form of precipitates, but a small amount will still dissolve in the matrix, leading to preferred orientation in the (100) direction, greater magneto-crystalline anisotropy and a greater tetragonal distortion range, which enhances the magnetostrictive property [27]. At 2.5–4.5%, λs reduces continuously, which is the result of the joint action of the increase in the lattice constant and the decrease in I_200_/I_110_ (%) and is consistent with the previous theory.

### 3.4. Microhardness

Figure 8 presents the curve of microhardness of the Fe_73_Ga_27−x_Al_x_ alloy and the Fe_73_Ga_27−x_Al_x_)_99.9_La_0.1_ alloy varying with Al content. Figure 8a shows that the hardness of the Fe_73_Ga_27−x_Al_x_ alloy increases first and then decreases with the change of Al content. In comparison to the Fe_73_Ga_27_ alloy, after the addition of the Al element, the hardness values of the Fe_73_Ga_27−x_Al_x_ alloy rise, which is related to the increase in ductility and plasticity after adding the Al element to the alloy [28]. The microhardness of the alloy measures up to the maximum value 273.183 HV at 2.5%. Compared with the alloys of undoped Al, although the hardness of the alloy increases by 6.6%, the λs decreases by 10.5%. While the λs of the alloy at x = 3.5% decreases by 2.6%, the hardness value increases by 4.6%. Therefore, by replacing the Ga element with appropriate Al elements in practical production, the Fe-Ga alloy device can obtain more economic benefits and excellent comprehensive properties. In light of Figure 8b, the hardness value of the alloy of x = 2.5% is up to the crest value 278.096 HV, which is higher than any of the Fe_73_Ga_27−x_Al_x_ alloys in accordance with the previous prediction results, suggesting that the appropriate composition of the La element doped into the Fe-Ga alloy can play a role in refining the grain and increasing the hardness of the alloy.

## 4. Conclusions

The Fe_73_Ga_27−x_Al_x_ alloy and Fe_73_Ga_27−x_Al_x_)_99.9_La_0.1_ alloys are still dominated by the A2 phase, but its diffraction peak position has a small angle deviation in comparison to the standard diffraction spectrum of α-Fe. After Al and La doping, the lattice constant and I_200_/I_110_ ratio of the alloy change, resulting in lattice distortion, further affecting the magnetostriction properties of the alloy.

The grains of the Fe_73_Ga_27−x_Al_x_ and Fe_73_Ga_27−x_Al_x_)_99.9_La_0.1_ alloys appear as distinct columnar crystals with certain orientations along the water-cooled direction. Moreover, the grains of the Fe_73_Ga_27−x_Al_x_)_99.9_La_0.1_ alloy are finer, exhibiting higher hardness values. It can be further speculated that the other mechanical properties of the alloys will be more excellent. EDS analysis shows that the Al element partially dissolves in the Fe-Ga solid solution, while the La element hardly gets inside the Fe-Ga alloy matrix, with masses of them existing in the form of the precipitation phase.

The saturation magnetostrictive strains of the Fe_73_Ga_27−x_Al_x_ alloy are between 58–76 ppm. Doping Al with proper composition can not only reduce the production cost, but also improve the hardness and magnetostriction properties. The saturation magnetostrictive strains of the Fe_73_Ga_27−x_Al_x_)_99.9_La_0.1_ alloy is in the range of 74–115 ppm, which proves that a small amount of La doping can wholly enhance the saturation magnetostrictive strain of Fe-Ga-Al. The hardness (278.096 HV) and the maximum saturation magnetostrictive strain (115 ppm) of the alloy at x = 2.5% are higher than those of other alloys, indicating that La doping can effectively meliorate the saturation magnetostrictive coefficient of the alloy and augment the hardness of the alloy.

## Figures and Tables

**Figure 1 materials-16-00012-f001:**
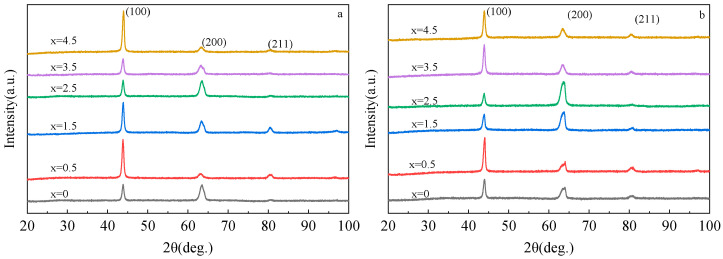
X-ray diffraction patterns of Fe_73_Ga_27−x_A_lx_ and Fe_73_Ga_27−x_Al_x_)_99.9_La_0.1_ alloys; (**a**) Fe_73_Ga_27−x_A_lx_;(**b**) Fe_73_Ga_27−x_Al_x_)_99.9_La_0.1_.

**Figure 2 materials-16-00012-f002:**
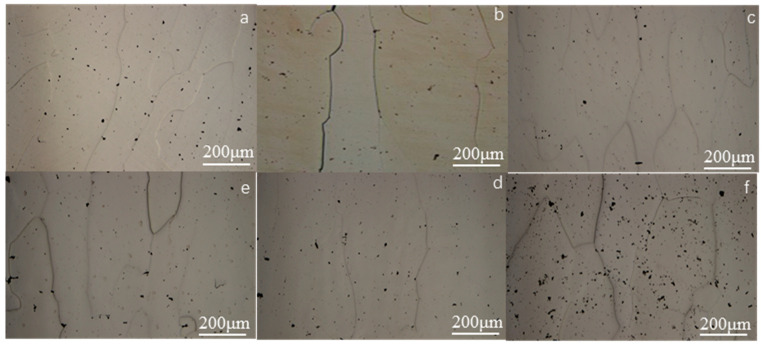
The metallographic photos of Fe_73_Ga_27−x_Al_x_ alloy (**a**) x = 0, (**b**) x = 0.5, (**c**) x = 1.5, (**d**) x = 2.5, (**e**) x = 3.5, (**f**) x = 4.5.

**Figure 3 materials-16-00012-f003:**
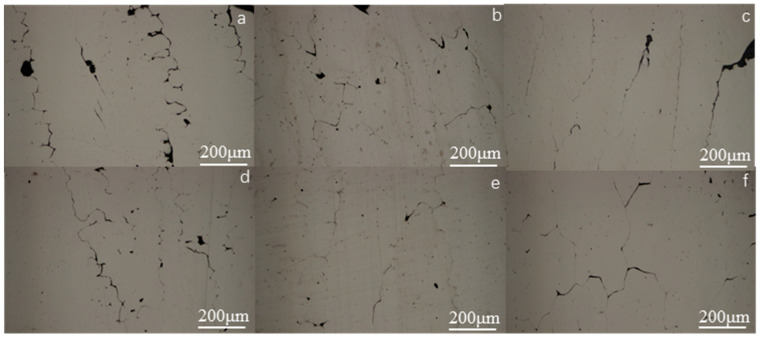
The metallographic photos of (Fe_73_Ga_27−x_Al_x_)_99.9_La_0.1_ alloy (**a**) x = 0, (**b**) x = 0.5, (**c**) x = 1.5, (**d**) x = 2.5, (**e**) x = 3.5, (**f**) x = 4.5.

**Figure 4 materials-16-00012-f004:**
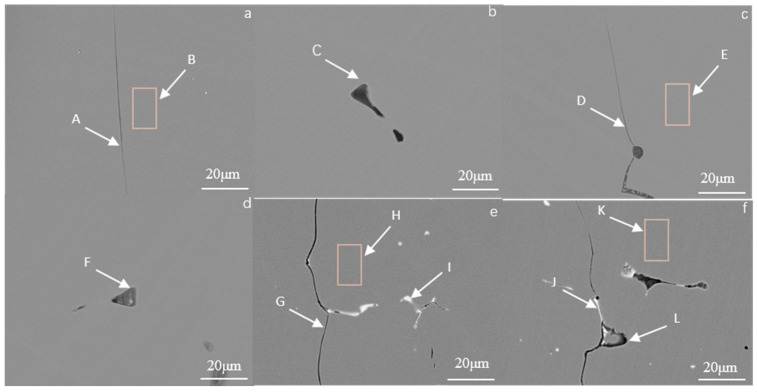
(**a**) x = 0, (**b**) x = 0, (**c**) x = 2.5, (**d**) x = 2.5 are SEM images of Fe_73_Ga_27−x_Al_x_ alloy; (**e**) x = 0, (**f**) x = 2.5 are SEM images of (Fe_73_Ga_27−x_Alx)_99.9_La_0.1_ alloy.

**Figure 5 materials-16-00012-f005:**
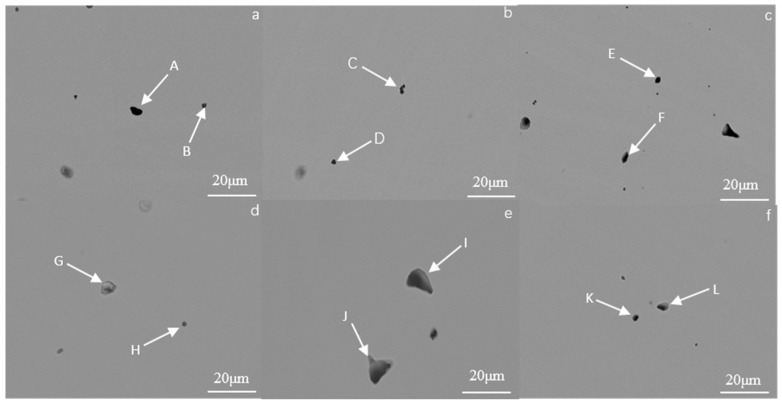
The black dots tissue of Fe_73_Ga_27−x_Al_x_ alloy ((**a**) x = 0, (**b**) x = 0.5, (**c**) x = 1.5, (**d**) x = 2.5, (**e**) x = 3.5, (**f**) x = 4.5) observed by SEM.

**Figure 6 materials-16-00012-f006:**
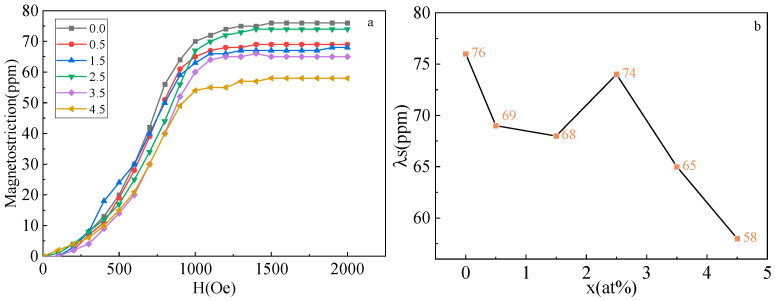
(**a**) Magnetostrictive curves of Fe_73_Ga_27−x_Al_x_ alloy; (**b**) Saturation magnetostrictive strain curve of alloys.

**Figure 7 materials-16-00012-f007:**
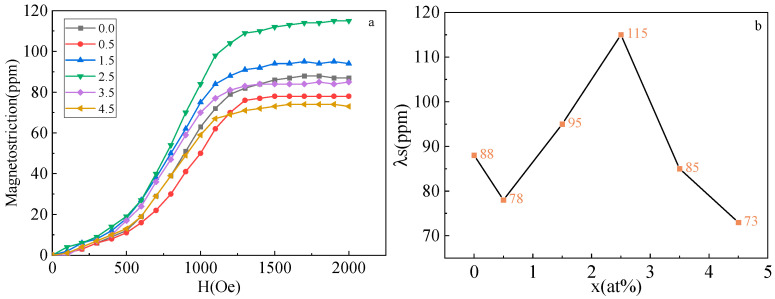
(**a**) Magnetostrictive curves of Fe_73_Ga_27−x_Al_x_)_99.9_La_0.1_ alloy; (**b**) Saturation magnetostrictive strain curve of alloys.

**Figure 8 materials-16-00012-f008:**
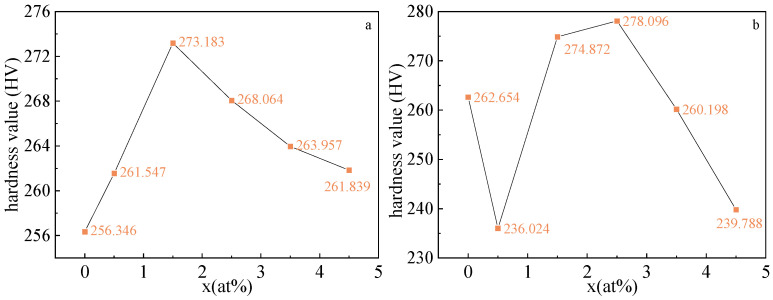
Microhardness (**a**) Fe_73_Ga_27−x_Al_x_; (**b**) Fe_73_Ga_27−x_Al_x_)_99.9_La_0.1_.

**Table 1 materials-16-00012-t001:** Lattice constants a(Å) and relative peak intensities (I_200_/I_110_) obtained by calculating XRD patterns of Fe_73_Ga_27−x_Al_x_ alloy and Fe_73_Ga_27−x_Al_x_)_99.9_La_0.1_ alloy.

Types of Alloys	Fe_73_Ga_27−x_Al_x_		(Fe_73_Ga_27−x_Al_x_)_99.9_La_0.1_	
x(%)	a(Å)	I_200_/I_110_(%)	a(Å)	I_200_/I_110_(%)
0	2.9251	83.8	2.9142	42.8
0.5	2.9263	9.3	2.9155	22.1
1.5	2.9289	33.1	2.9157	89.0
2.5	2.9203	80.4	2.9132	140.77
3.5	2.9215	46.7	2.9184	28.6
4.5	2.9243	10.0	2.9226	26.9

**Table 2 materials-16-00012-t002:** EDS analysis results of Fe_73_Ga_27−x_Al_x_ and (Fe_73_Ga_27−x_Al_x_)_99.9_La_0.1_ alloys in different regions.

Micro-Zones		Fe_73_Ga_27−x_Al_x_			Micro-Zones		(Fe_73_Ga_27−x_Al_x_)_99.9_La_0.1_			
		Fe	Ga	Al			Fe	Ga	Al	La
x = 0	A	69.33	30.67	0	x = 0	G	50.39	39.65	0	9.96
	B	73.87	26.13	0		H	75.25	24.69	0	0.06
	C	77.56	22.44	0		I	36.92	47.78	0	15.30
x = 2.5	D	72.02	27.52	0.46	x = 2.5	J	74.99	23.87	1.10	0.05
	E	72.22	27.06	0.72		K	74.77	23.99	2.19	0.06
	F	80.06	18.52	1.42		L	17.68	58.13	0.44	23.75

**Table 3 materials-16-00012-t003:** EDS analysis of black dots tissue of Fe_73_Ga_27−x_Al_x_ alloy (x = 0, 0.5, 1.5, 2.5, 3.5, 4.5) alloy.

Micro-Zones		Fe_73_Ga_27−x_Al_x_			Micro-Zones		Fe_73_Ga_27−x_Al_x_		
		Fe	Ga	Al			Fe	Ga	Al
x = 0	A	83.45	16.55	0	x = 2.5	G	71.75	26.71	1.53
	B	73.87	26.13	0		H	71.94	25.95	2.11
x = 0.5	C	29.81	10.30	59.89	x = 3.5	I	77.77	22.10	0.13
	D	30.39	9.16	60.45		J	78.24	21.76	0
x = 1.5	E	76.82	23.18	0	x = 4.5	K	34.25	10.07	55.68
	F	77.08	22.92	0		L	72.54	24.81	2.65

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
