# Peer review of "Effect of Al and La Doping on the Structure and Magnetostrictive Properties of Fe73Ga27 Alloy"

_materials, 2022, doi:10.3390/ma16010012_

Round 1

Reviewer 1 Report

The manuscript provides a well-detailed research on FeGaAl and FeGaAlLa alloys, with different Al contents, where the structure, the magnetic properties and the hardness were investigated. These alloys have promising properties, thus this research is profitable and the manuscript contains useful knowledge. 

The manuscript is well-written, the introduction and the discussion are fine, the methods are described very well, and also the conclusions are supported by the results. I accept it for publication after a few very minor modifications:

- There is a misprint in the caption of table 1: the lattice parameter "a" should be in minuscule. Also, it would be good to indicate in the table which results are belonging to FeGaAl and which results are belonging to FeGaAlLa

-There are also misprints in the text, like "Maintai" in line 81 or "Moreove" in line 278.

Author Response

Thank you for your comments. I have revised the manuscript according to your requirements.

Reviewer 2 Report

The article:

Effect of Al and La Doping on the Structure and Magnetostric- tive Properties of Fe73Ga27 Alloy” ,

 by: Jinchao Du, Pei Gong, Xiao Li, Shaoqi Ning, Wei Song, Yuan Wang , Hongbo Hao,

deals with interesting approach to experimental studies of  the changes of microstructure, magnetostriction properties and hardness of Fe73Ga27-xAlx  alloy and (Fe73Ga27-xAlx)99.9La0.1 alloy (x=0, 0.5, 1.5, 2.5, 3.5, 4.5) by doping Al and La into Fe73Ga27 and (Fe73Ga27)99.9La0.1 alloys respectively. In order to obtain the appropriate results, the authors used for the research following methods:  X-ray diffraction, SEM micrographs, and Magnetic measurements. In conclusion the authors concluded that the saturation magnetostrictive strains of Fe73Ga27-xAlx alloy is between 58-76ppm.  Doping Al with proper composition can not only reduce the production cost, but also give  consideration to the hardness and magnetostriction properties.

I have gone through the manuscript and I recommend it for publication in Materials with the following reasons:

The paper is scientifically sound and of important interest.

The information presented in the paper is original and comprehensive.

The results and discussion sections were well-written and thorough.

Although the article is very interesting but before publishing requires a following addition:

1. What is the order of magnitude of the measurement errors presented, among others, in Figures 5 and 6.

2. Please correct the markings on the Y-axis in Figure 7.

3. Taking into account the results of experiments and calculations the authors presented too modest interpretation in the "Conclusion". I propose to expand more boldly applications especially in practical use.

Author Response

Thank you for your comments. I have tried my best to revise the manuscript according to your requirements.

  1. What is the order of magnitude of the measurement errors presented, among others, in Figures 5 and 6.

The resistance strain gauge measurement method used in the experiment has an error range of 5ppm.

  1. Please correct the markings on the Y-axis in Figure 7.

Figure 7 has been revised as follows:

  1. Taking into account the results of experiments and calculations the authors presented too modest interpretation in the "Conclusion". I propose to expand more boldly applications especially in practical use.

Thank you for affirming the conclusion of the manuscript, but this is just the work in the preliminary theoretical research. When it comes to the actual production application, it is necessary to consider the specific application direction, which existing devices can be replaced by the alloy, which new devices can be produced, as well as their service environment, service life, and other properties. If possible, I will continue to study their application in actual production and life.

If there are any questions in the manuscript, please let me know.

Reviewer 3 Report

I encourage the authors to write such an article and do such valuable research. I believe that this manuscript can be published in the present form. in other words, accept. 

One of the main advantage of this research compared to other is to describe materials exactly and in details. In addition, the obtained results were well interpreted.

Author Response

Thank you for your comments and support for the content of the manuscript. If there are any questions in the manuscript, please let me know.

Reviewer 4 Report

In the manuscript entitled “Effect of Al and La Doping on the Structure and Magnetostrictive Properties of Fe73Ga27 alloy” authors study the changes of microstructure, magnetostriction properties and hardness of Fe73Ga27-xAlx and (Fe73Ga27-xAlx)99.9La0.1 (x=0, 0.5, 1.5, 2.5, 3.5) alloys. The results could be interesting and important for the possible applications of these compounds as magnetostrictive material due to the low saturated magnetic field, good mechanical properties and low material cost that they exhibit. However, the following issues have to be addressed before considering the manuscript for publication:

1.       Typos. Some examples: Which (line 25), curie (line 31), visbly (152), moreove (278)

2.       Please, specify how the samples were prepared. It is not clear in section 2. Arc-melting?

3.       Please, explain which is the meaning of good shape (line 83)

4.       Please, include some words explaining how the samples were prepared for the observation by SEM. Indicate if the XRD were performed in bulk or powder samples.

5.       Figure 1. Please, indicate how the lattice constant has been calculated. It would be interesting to carry out a fitting of the XRD patterns (LeBail or Rietvield) in order to obtain more accurate values.

6.       Table 1. I think that is more visible substitute this table for a figure showing the evolution of the lattice constants and relative peak intensities. In the way in which it is shown, it is not easy to follow. Please, include errors.

7.       Figure 2. Have the authors considered the possible existence of pores in the samples? Which is the meaning of columnar crystals (line 164)? In which direction? The same for Figure 3.

8.       Figure 4. Quality of the figure should be improved. Which is the spot size for the EDS analysis?

9.       Table 2. Please, include errors estimation and the values obtained for the other compositions. It is clear observed than some zones are richer in Ga. The experimental mean composition of the samples should be estimated by other methods, as XRF.

9.    Please, include some discussion of Figs. 6 and 7 about the values obtained in the literature for the magnetostriction of Fe-Ga compounds.

9.    Figure 7. Axis y in English. Indicate errors. How many measurements have been performed in order to determine the hardness of the samples?

Author Response

Thank you for your comments. I have tried my best to revise the manuscript according to your requirements.

  1. Some examples: Which (line 25), curie (line 31), visbly (152), moreove (278)

I checked the article again to minimize spelling mistakes. I have corrected all the errors you pointed out.

  1. Please, specify how the samples were prepared. It is not clear in section 2. Arc-melting?

All samples were prepared by vacuum arc melting furnace.

  1. Please, explain which is the meaning of good shape (line 83)

A “good shape” alloy is usually distinguished by the shape of alloys, whether the surface is oxidized, whether the massive defects appear in the surface.

  • The alloy melted in the vacuum smelting furnace is button shaped under the cooling of the water-cooled copper crucible.
  • If the alloy’s surface is not oxidized, the color of the surface is gray and otherwise the alloy’s surface is black, or a fraction is colorful.
  • The fewer defects on the surface of alloy indicates that the better the alloy is smelted and the more homogeneous the composition is.
  1. Please, include some words explaining how the samples were prepared for the observation by SEM. Indicate if the XRD were performed in bulk or powder samples.
  • A 12*12*3 mm sheet or a 12*12*10 mm block is cutted from the melted alloy by linear cutting.
  • The use of 320 #, 500 #, 800 #, 1000 #, 1500 #, 2000 #, 2500 # sandpaper in the polishing machine for grinding, the sample should be rotated 90 degrees when replaced abrasive paper ench time, to ensure that the scratch can cover the last scratch.
  • When the scratch is tiny and shallow, use 0.5 μm polishing paste on the polishing machine to polish until the surface of the sample is as bright as a mirror.
  • Under SEM, the grain boundaries and grains of the alloy are observed. If they are not visible, the observation surface of the sample can be corroded with 4% nitrate alcohol solution.

The XRD sample used in this experiment is bulk and their size is 12*12*10 mm.

  1. Figure 1. Please, indicate how the lattice constant has been calculated. It would be interesting to carry out a fitting of the XRD patterns (LeBail or Rietvield) in order to obtain more accurate values.

The lattice constants of the alloys were calculated using MDI Jade software.The specific steps include importing the measured XRD data into Jade software firstly, then calibrating the main peaks, removing the stray peaks, and finally calculating the lattice constant of the alloy through the “report”function.

  1. Table 1. I think that is more visible substitute this table for a figure showing the evolution of the lattice constants and relative peak intensities. In the way in which it is shown, it is not easy to follow. Please, include errors.

The table 1 has been revised as follows:

Types of alloys

Fe73Ga27-xAlx

(Fe73Ga27-xAlx)99.9La0.1

x(at%)

a(Å)

I200/I110(%)

a(Å)

I200/I110(%)

0

2.9251

83.8

2.9142

42.8

0.5

2.9263

9.3

2.9155

22.1

1.5

2.9289

33.1

2.9157

89.0

2.5

2.9203

80.4

2.9132

140.77

3.5

2.9215

46.7

2.9184

28.6

4.5

2.9243

10.0

2.9226

26.9

  1. Figure 2. Have the authors considered the possible existence of pores in the samples? Which is the meaning of columnar crystals (line 164)? In which direction? The same for Figure 3.

During the casting process, the pores are existed in the samples inevitably. We only need to ensure that the pores on the tested side of the sample are as small as possible. Most of the pores situated on the surface can be burnished through abrasive paper.

The “columnar crystals” refers to the crystal belt with distinct directional growth facing heat flow after the formation of fine equiaxed crystal, which appears as columnar, so the crystal zone is called columnar crystal.

In this experiment, the columnar crystals grow along the cooling direction of the water-cooled copper crucible

  1. Figure 4. Quality of the figure should be improved. Which is the spot size for the EDS analysis?

Quality of the figure 4 has been improved as follows:

The spot size for the EDS analysis is 0.15 um.

  1. Table 2. Please, include errors estimation and the values obtained for the other compositions. It is clear observed than some zones are richer in Ga. The experimental mean composition of the samples should be estimated by other methods, as XRF.

According to previous experience, the error estimation of EDS analysis by SEM usually fluctuates within 1%.

In the light of the results in the next figure and table , the composition of large proportion of the black dots is similar to the composition of their respective matrix, and observing the points C, D and K, the relative contents of Al elements were 59.89, 60.45 and 55.68, re-spectively, which were higher than those of Fe and Ga, and no Ga-rich region was ob-served in the alloy, so we can speculate that the black dots tissue in the alloy is more likely to be composed of the deciduous matrix phase and Al element in the form of in-clusion.

Micro-zones

Fe73Ga27-xAlx

Micro-zones

Fe73Ga27-xAlx

Fe

Ga

Al

Fe

Ga

Al

x=0

A

83.45

16.55

0

x=2.5

G

71.75

26.71

1.53

B

73.87

26.13

0

H

71.94

25.95

2.11

x=0.5

C

29.81

10.30

59.89

x=3.5

I

77.77

22.10

0.13

D

30.39

9.16

60.45

J

78.24

21.76

0

x=1.5

E

76.82

23.18

0

x=4.5

K

34.25

10.07

55.68

F

77.08

22.92

0

L

72.54

24.81

2.65

I'm really sorry that XRF analysis cannot be performed temporarily due to the fragmentary preservation of sample and the dearth of experimental equipment in the school.

  1. Please, include some discussion of Figs. 6 and 7 about the values obtained in the literature for the magnetostriction of Fe-Ga compounds.

Some discussion of Figs 6 and 7 about the values obtained in the literature has been added.

  1.  Figure 7. Axis y in English. Indicate errors. How many measurements have been performed in order to determine the hardness of the samples?

Figure 7 has been revised as follows:

During the experiment, the force exerted by the indenter on the sample is 200 gf(1.962 N), and the load is kept for 10 s. At least 5 points are hit on each sample and the average value of its effective value is calculated, namely the microhardness value of the sample.

If there are any questions in the manuscript, please let me know. I’m looking forward to hearing from you.

Round 2

Reviewer 4 Report

The comments have been suscessfully answered. I recommend the publication of the manuscript in the present form